# Predictors and outcome of first line treatment failure among under-five children with community acquired severe pneumonia at Bugando Medical Centre, Mwanza, Tanzania: A prospective cohort study

**Restituta Phabian Muro[1], Tulla Sylvester Masoza[1,2]\*, Godfrey Kasanga[3], Neema Kayange[1,2], Benson R. Kidenya[4]**

1 Department of Pediatrics and Child Health, Bugando Medical Centre, Mwanza, Tanzania, 2 Department of Pediatrics and Child Health, Catholic University of Health and Allied Sciences–Bugando, Mwanza, Tanzania, 3 Department of Radiology, Bugando Medical Centre, Mwanza, Tanzania, 4 Department of Biochemistry and Molecular Biology, Catholic University of Health and Allied Sciences–Bugando, Mwanza, Tanzania

\* sylvestertullah@gmail.com

## Abstract

### Background

Despite recent advances in management and preventive strategies, high rates of first line antibiotics treatment failure and case fatality for Severe Community Acquired Pneumonia (SCAP) continue to occur in children in low and middle-income countries. This study aimed to identify the predictors and outcome of first line antibiotics treatment failure among children under-five years of age with SCAP admitted at Bugando Medical Centre (BMC) in Mwanza, Tanzania.

### Methods

The study involved under-five children admitted with SCAP, treated with first line antibiotics as recommended by WHO. Patients with treatment failure at 48 hours were shifted to second line of antibiotics treatment and followed up for 7 days. Generalized linear model was used to determine predictors of first line antibiotics treatment failure for SCAP.

### Results

A total of 250 children with SCAP with a median age of 18 [IQR 9–36] months were enrolled, 8.4% had HIV infection and 28% had acute malnutrition. The percentage of first line antibiotics treatment failure for the children with SCAP was 50.4%. Predictors of first line treatment failure were; presentation with convulsion (RR 1.55; 95% CI [1.11–2.16]; p-value 0.009), central cyanosis (RR 1.55; 95% CI [1.16–2.07]; p-value 0.003), low oxygen saturation (RR 1.28; 95% CI [1.01–1.62]; p-value 0.04), abnormal chest X-ray (RR 1.71; 95% CI [1.28–2.29]; p-value <0.001), HIV infection (RR 1.80; 95% CI [1.42–2.27]; p-value 0.001), moderate acute malnutrition (RR 1.48; 95% CI [1.04–2.12]; p-value = 0.030) and severe acute

**Data Availability Statement:** All relevant data are within the manusript and its supporting information files.

**Funding:** The authors received no specific funding for this work.

**Competing interests:** The authors declared no competing interests exist.

malnutrition (RR 2.02; 95% CI [1.56–2.61]; p-value<0.001). Mortality in children who failed first line treatment was 4.8%.

## Conclusion

Half of the children with SCAP at this tertiary center had first line antibiotics treatment failure. HIV infection, acute malnutrition, low oxygen saturation, convulsions, central cyanosis, and abnormal chest X-ray were independently predictive of first line treatment failure. We recommend consideration of second line treatment and clinical trials for patients with SCAP to reduce associated morbidity and mortality.

## Introduction

Pneumonia is one of the leading causes of morbidity and mortality in children under five years of age, causing about 1.6 million deaths every year globally [1, 2]. Globally, there is an estimated incidence of 151 million new cases of childhood pneumonia each year, and 7% - 13% in developing countries are severe enough to require hospitalization [1, 3, 4]. At Bugando Medical Centre (BMC), childhood pneumonia burden is 10.2% and is among the most common causes of pediatric admission contributing to high rates of morbidity and mortality. As childhood pneumonia is associated with significant mortality among children under five years, the World Health Organization (WHO) has developed standard case management (SCM) guidelines to reduce pneumonia-related mortality, using clinical signs to assess for pneumonia [5]. The SCM approach has been effective in reducing childhood pneumonia-related mortality by 50% and overall child mortality by 25% [5]. The current treatment strategies for pediatric pneumonia have been affected by the development of antibiotics resistance among pneumococcal and *Staphylococcus. aureus* strains [6, 7]. The 2013 WHO guidelines recommend ampicillin and gentamicin as first line antibiotics choice for all children with severe pneumonia of less than 5 years of age [8]. The "48 hours rule" has been included in the guidelines; if symptoms and signs of pneumonia have not started to improve at 48 hours, the child must be re-evaluated and alternative second line treatment should be initiated and the child is said to have first line treatment failure [9, 10].

There is a limited understanding of what clinical symptoms and signs may predict illness progression that warrants a change of first line antibiotics treatment to second line antibiotics among children with severe pneumonia. Identification of predictors that could potentially be responsible for the treatment failure at baseline is of paramount importance to guide when deciding for proper and more effective management of severe pneumonia in a child, which consequently may shorten the hospital stay and reduce management costs. Therefore, this study aimed at identifying the predictors of first line treatment failure among patients under the age of five years with severe pneumonia at Bugando Medical Centre.

## Materials and methods

This was a prospective cohort study with seven days follow up, involving children aged 2 to 59 months admitted with SCAP in pediatric wards of BMC from October 2013 to February 2014. BMC is a zonal referral hospital as well as a consultant and teaching hospital, located in the city of Mwanza, the second largest city in Tanzania. The hospital serves a population of approximately 16 million people from six regions in North-western Tanzania. The pediatrics

and child health department has a bed capacity of 89 and an average of 2–3 children with severe pneumonia are admitted at BMC every day. The research was approved for ethical clearance by the joint CUHAS-Bugando/BMC Ethical and Research review Committee board and was approved for ethical clearance and parents/guardians signed an informed consent form before enrolment of their children in our study. In this study diagnosis of SCAP was adopted from WHO pocketbook guideline of 2013 [8] whereby it warranted the presence of cough or difficulty in breathing, plus at least one of the following: Central cyanosis or oxygen saturation < 90% on pulse oximetry, severe respiratory distress, signs of pneumonia with a general danger sign such as inability to breastfeed or drink, lethargy or unconscious, convulsions. Whereby, in addition, some or all of the other signs of pneumonia may be present, such as fast breathing rate per age (age 2–11 months, ≥ 50/min age 1–5 years, ≥ 40/min), lower chest wall indrawing, chest auscultation signs e.g. decreased breath sounds, bronchial breath sounds, crackles, abnormal vocal resonance and pleural rub. Therefore admitted children with SCAP who were initiated on first line antibiotics treatment and whom their caregiver had given consent were recruited in the study serially until when the sample size was reached. Exclusion criteria included children with SCAP who were kept on second line antibiotics therapy during admission or who were on any antibiotics therapy for over 48 hours before admission, children who were on anti TB treatment, children with asthma and children with a congenital heart disease or child with heart murmur with evidenced cardiac pathology, screened by electrocardiogram or echocardiography examinations.

After obtaining consent, baseline clinical assessment, chest radiograph and laboratory specimens for blood smear for malaria parasite were obtained within 1 hour after enrolment. Vital signs, oxygen saturation, clinical signs and impression were recorded 6-hourly. Indications for providing oxygen, antipyretics, and bronchodilator treatment were standardized across the study site. Supportive therapies were given as needed after vital signs are assessed and recorded for each time interval. The caretakers were also offered pre-and post-counseling for HIV testing. The principal investigator and research assistants ensured that the study patients received the appropriate treatment and supportive care as prescribed by the clinician. Patients were followed up for seven days or until discharge (whichever was shorter) and outcomes were recorded at discharge or on day 7 of follow up, as still in the ward or dead.

First line treatment failure for SCAP was defined as a failure to improve or have normalization of the respiratory rate, oxygen saturation, and temperature, along with the presence of pulmonary complications and/or presence of clinical danger signs (severe chest wall in drawing, grunting, inability to breastfeed or drink, lethargy, and convulsion) or death up to 48 hours while the patient is on appropriate first line antibiotics treatment [8, 11, 12]

## Sample collection and laboratory procedures

A blood smear for malaria parasite was done using Giemsa stain (Ranbaxy, India) and the number of asexual parasites/200WBC was reported after one hour. HIV rapid test was done using finger prick blood samples in the wards by a qualified counselor. A rapid antibody test was used to screen HIV infection in all patients aged above 18 months. Determine HIV1/2 manufactured by Alere Medical Co. Ltd Japan was used as the first antibody test and Unigold manufactured by Trinity Biotech Plc, Bray, Ireland was used as the second confirmatory antibody test for those with positive determine HIV test. DNA Polymerase Chain Reaction (PCR) technique for detection of HIV genetic material was used to screen infants and children under the age of 18 months.

A standard antero-posterior chest radiograph was obtained for every child. Two independent experts (consultant pediatricians), blinded to all clinical information, evaluated each

radiograph and documented their findings on a standard report form. Where the two objective experts disagreed, a third expert (radiologist) made the final decision.

**Statistical analysis.**   Data analysis was done using STATA software version 13.0 (College Station, Stata Corp, Texas, USA). Data were summarized in the form of frequency tables using percentages for categorical variables. Median with Interquartile range (IQR) was used to summarize continuous variables. The primary outcome of interest was first line antibiotics treatment failure for SCAP that was observed at 48 hours after initiation of treatment. Generalized linear model with log link and a Poisson distribution with robust variance estimator was used to determine predictors of first line antibiotics treatment failure of SCAP. This model was used to eliminate the convergence faced when using the log-binomial model [13]. Relative Risk with 95% confidence interval (CI) was calculated to measure the strength of association between predictor variables and outcome (first line antibiotics treatment failure). Predictors with a p-value of <0.2 when calculating for unadjusted RR were further analyzed to adjust for the confounding effect. Predictors with a p-value of less than 0.05 after calculating for adjusted RR were considered as significant independent predictors of first line antibiotics treatment failure.

## Results

### Participants' enrolment and characteristics

During the study period, a total of 978 children were admitted at BMC pediatrics wards. Seven hundred and eighteen (718) children did not meet the eligibility criteria whereby, 416 children were out of study age limit, 269 children had no features of SCAP, and 33 children with SCAP were started on second line antibiotics treatment during admission. Two hundred and sixty (260) children aged 2 to 59 months were eligible for enrolment, out of which ten (3.8%) of them had no parental consent. Finally, 250 children were included in the final analysis as shown in "Fig 1".

The median age of the enrolled children was 18 [IQR 9–36] months. Of the 250 children, 145 (58.0%) were males and 144 (57.6%) were below 2 years of age. All of the 250 children had a cough, difficulty in breathing; higher respiratory rate, nasal flaring and lower chest wall in drawing. A majority of these children, 241 (96.4%) were lethargic, 228 (91.2%) had crackles, 202 (80.8%) were unable to breastfeed and drink, while 164 (65.6%) had temperature ≥38°C, 81 (32.4%) had oxygen saturation of < 90%, 32 (14.4%) had central cyanosis, 59 (23.6%) vomited everything, 18 (7.2%) had convulsions, and 126 (51.6%) had respiratory distress. Respiratory distress in this study context was considered as the presence of grunting and head nodding. There were 70 (28.0%) children with malnutrition, 21 (8.4%) with HIV infection and 15 (6.0%) with malaria. In this study, an acute form of malnutrition was assessed among the participants using the WHO Z-score charts of weight for height or length. Z-scores of between -3SD and ≤ -2SD were equated to moderate acute malnutrition and Z-scores of ≤ -3SD were equated to severe acute malnutrition, also the presence of kwashiorkor was considered as severe acute malnutrition. "Table 1" summarizes the distribution of demographic and clinical characteristics of children with SCAP.

### Proportion, predictors and outcome of first line antibiotic treatment failure

First line antibiotics (Ampicillin and Gentamicin) treatment failure was observed in 50.4% (126/250) of children with SCAP. These children were shifted to second line antibiotic treatment which was ceftriaxone. After analyzing for adjusting RR. presentation with the following symptoms and signs were predictive of treatment failure in SCAP; convulsions (RR 1.55; 95% CI [1.11–2.16]; p-value 0.009), central cyanosis (RR 1.55; 95% CI [1.16–2.07]; p-value 0.003)

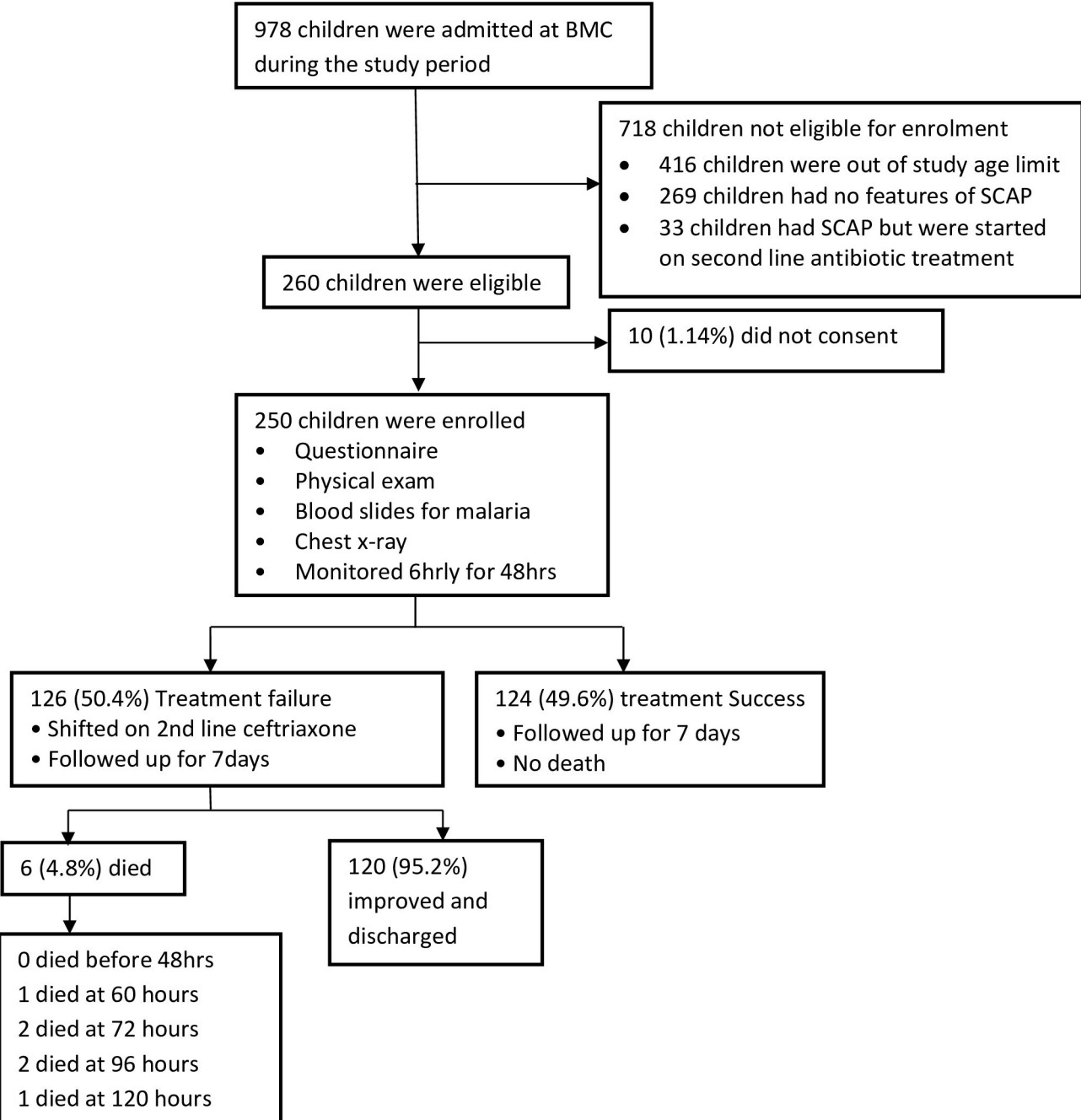

**Fig 1. A flow diagram displaying participants' enrolment and outcome.**

and low oxygen saturation (RR 1.28; 95% CI [1.01–1.62]; p-value 0.04). Moreover co-existence of SCAP with HIV infection (RR 1.80; 95% CI [1.42–2.27]; p-value <0.001), moderate acute malnutrition (RR 1.48; 95% CI [1.04–2.12]; p-value 0.03) and severe acute malnutrition (RR 2.02; 95% CI [1.56–2.61]; p-value<0.001) was found to predict first line treatment failure for SCAP as depicted in "**Table 2**". After seven days follow up of 126 children who were shifted to second line treatment, 6 (4.8%) had died. Out those who died 3 (50.0%) had SCAP alone, 2 (33.3%) had severe malnutrition and 1 (16.7%) had HIV infection.

**Table 1. Demographic information and clinical characteristics among 250 children with severe community acquired pneumonia admitted at BMC from October 2013 to February 2014.**

| Symptoms/Signs | Number | Percent (%) |
|---|---|---|
| *Male sex* | 145 | 58.0 |
| *Age < 2 years* | 144 | 57.6 |
| *Unable to drink or breastfeed* | 202 | 80.8 |
| *Convulsion* | 18 | 7.2 |
| *Vomiting everything* | 59 | 23.6 |
| *Body temperature >38.0°C* | 164 | 65.6 |
| *Lethargy* | 241 | 96.4 |
| *Central cyanosis* | 36 | 14.4 |
| *Bronchial Breathing Sound +/- Crackles* | 228 | 91.2 |
| *Respiratory distress* | 129 | 51.6 |
| *Oxygen saturation < 90%* | 81 | 32.4 |
| *Moderate acute malnutrition* | 24 | 9.6 |
| *Severe acute malnutrition* | 46 | 18.4 |
| *HIV positive* | 21 | 8.4 |
| *Positive blood slide for malaria* | 15 | 6.0 |

**Chest X-ray findings in 250 children admitted in pediatric wards with SCAP.** The Chest X–ray was done in all 250 children with SCAP. Of these, 146 (58.4%) had features suggestive of pneumonia such as bronchopneumonia, multilobe pneumonia and lobar pneumonia. The remaining, 104 (41.6%) children had normal chest X-ray findings. Treatment failure was more observed in children with abnormal chest X-ray than in those with normal chest X-ray (61.0% [89/146] versus 35.6% [37/104]); (RR 1.71; 95% CI [1.28–2.29]; p-value <0.001).

## Discussion

### The proportion of patients with first line antibiotics treatment failure

Half (50.4%) of all participants failed first line antibiotic treatment for SCAP. A figure similar to 56.5% documented in a study done in New Delhi India which reported treatment failure for Amipicilin and Gentamicin in SCAP (9) but higher than 20% and 12.5% reported in a study done in Kenya [10] and South Africa [14] respectively. The high proportion of treatment failure in our study could be due to several reasons. First, there is a possibility of resistant strain to first line antibiotics used in the treatment of SCAP. We were unable to determine whether most cases in those that failed therapy were due to bacterial or viral causes. However, WHO has recommended the initiation of antibiotics for all children presenting with a clinical syndrome consistent with SCAP. Second, children presented in a late stage of the disease that prevented quick recovery within the first 48 hours since the study site was a tertiary hospital hence a high likelihood of receiving late referrals. However, most severe bacterial infections improve in the first 48 hours on parenteral antibiotics unless there is septic shock or antimicrobial resistance. Third, poor response to first line treatment could be attributed by the presence of multiple comorbidities such as malnutrition and HIV infection. As a matter of fact, in our study, 8.4% of children with SCAP had HIV infection and 28.0% had moderate to severe acute malnutrition.

### The predictors of the first line treatment failure for SCAP

In this study, we found the independent predictors of first line treatment failure in children with SCAP to be HIV infection, moderate and severe acute malnutrition, low oxygen saturation,

**Table 2. Predictors of first line treatment failure among 250 children with severe community acquired pneumonia admitted at BMC from October 2013 to February 2014.**

| Patient's features | First line Treatment failure | | RR [95% CI] | P-value | Adjusted RR [95% CI] | P-value |
|---|---|---|---|---|---|---|
| | Yes | No | | | | |
| | n (%) | n (%) | | | | |
| *Age in years* | | | | | | |
| ≥ 2 | 45 (42.5) | 61 (57.5) | 1.0 | | 1.0 | |
| < 2 | 81 (56.3) | 63 (43.7) | 1.33[1.02–1.73] | 0.037 | 1.20[0.94–1.54] | 0.13 |
| *Sex* | | | | | | |
| Male | 71 (49.0) | 74 (51.0) | 1.0 | | | |
| Female | 55 (52.4) | 50 (47.6) | 1.06[0.84–1.37] | 0.593 | - | - |
| *Unable to breastfeed or drink* | | | | | | |
| No | 17 (35.4) | 31 (64.6) | 1.0 | | 1.0 | |
| Yes | 109 (54.0) | 93 (46.0) | 1.52[1.02–2.28] | 0.041 | 1.31[0.94–1.88] | 0.14 |
| *Convulsion* | | | | | | |
| No | 112 (48.3) | 120 (51.7) | 1.0 | | 1.0 | |
| Yes | 14 (77.8) | 4 (22.2) | 1.61[1.22–2.13] | 0.01 | 1.55[1.11–2.16] | 0.009 |
| *Fever ≥38°C* | | | | | | |
| No | 45 (52.3) | 41 (47.7) | 1.0 | | | |
| Yes | 81 (49.4) | 83 (50.6) | 0.94[0.73–1.22] | 0.656 | - | - |
| *Vomiting Everything* | | | | | | |
| No | 91 (47.6) | 100 (52.4) | 1.0 | | 1.0 | |
| Yes | 35 (59.3) | 24 (40.7) | 1.25[0.96–1.61] | 0.096 | 1.00[0.77–1.31] | 0.99 |
| *Lethargic* | | | | | | |
| No | 6 (66.7) | 3 (33.3) | 1.0 | | | |
| Yes | 120 (49.8) | 121 (50.2) | 0.75[0.46–1.21] | 0.232 | - | - |
| *Central cyanosis* | | | | | | |
| No | 98 (45.8) | 116 (54.2) | 1.0 | | 1.0 | |
| Yes | 28 (77.8) | 8 (22.2) | 1.70 [1.35–2.13] | <0.001 | 1.55[1.16–2.07] | 0.003 |
| *Respiratory distress* | | | | | | |
| No | 49 (40.5) | 72 (59.5) | 1.0 | | 1.0 | |
| Yes | 77 (59.7) | 52 (40.3) | 1.47 [1.14–1.91] | 0.003 | 1.11[0.85–1.44] | 0.44 |
| *Low Oxygen Saturation* | | | | | | |
| No | 73 (43.2) | 96 (56.8) | 1.0 | | 1.0 | |
| Yes | 53 (65.4) | 28 (34.6) | 1.51[1.20–1.91] | 0.001 | 1.28[1.01–1.62] | 0.04 |
| *HIV status* | | | | | | |
| Negative | 106 (46.3) | 123 (53.7) | 1.0 | | 1.0 | |
| Positive | 20 (95.2) | 1 (4.8) | 2.06 [1.74–2.44] | <0.001 | 1.80 [1.42–2.27] | <0.001 |
| *Nutritional status* | | | | | | |
| Normal | 73 (40.6) | 107 (59.4) | 1.0 | | 1.0 | |
| Moderate acute malnutrition | 15 (62.5) | 9 (37.5) | 0.43[0.08–0.79] | 0.018 | 1.48 [1.04–2.12] | 0.03 |
| Severe acute malnutrition | 38 (82.6) | 8 (17.4) | 0.71[0.49–0.93] | <0.001 | 2.02 [1.56–2.61] | <0.001 |
| *Malaria* | | | | | | |
| No | 118 (50.2) | 117 (49.8) | 1.0 | | | |
| Yes | 8 (53.3) | 7 (46.7) | 1.06[0.65–1.73] | 0.810 | - | - |

abnormal chest X-ray findings, central cyanosis, and convulsion. Studies that were done in Kenya [10] and South Africa [12] have reported similar effects of HIV infection on response to first line treatment for SCAP. HIV infection has probably changed the spectrum of pathogens

that cause childhood pneumonia presenting with recurrent bacterial pneumonia and possibly in severe forms with increased antimicrobial resistance of the common respiratory pathogens [15]. Development of antimicrobial resistance could partly be explained by the higher likelihood of frequent hospital treatments or over the counter treatment of respiratory tract infections with antibiotics and thus affecting the response to first line treatments used for severe pneumonia [15]. The finding that severe acute malnutrition was a predictor of first line treatment failure for SCAP has also been reported in other studies [10, 16]. Children with severe acute malnutrition present with subtle features of severe pneumonia and thus leading to a delay in initiating parenteral antibiotics [17]. Furthermore, there is a wide range of bacterial pathogens predominantly the gram-negative bacteria associated with severe pneumonia or bacteremia in children with severe acute malnutrition. Most of these gram-negative bacteria are resistant to ampicillin and gentamicin, the first line drugs used for the treatment of severe pneumonia or bacteremia among these children [18–20]. The study found a third of patients presented with oxygen saturation <90% and half of these children also had central cyanosis. Furthermore, children with low oxygen saturation were more likely to get first line antibiotics treatment failure than those with normal oxygen saturation. This could be explained by the fact that oxygen is needed by immunity cells to successfully combat pathogens [14, 21] and children who present in late stage usually have consolidated lung [22] that cause low oxygen exchange in the alveolus. Finally, in our study, 61.0% of children with abnormal chest X-ray findings had first line antibiotics treatment failure. The association of chest X-ray findings with treatment failure is likely since more virulent bacterial pathogens cause focal lung radiologic abnormalities [23].

## Outcome after first line antibiotics treatment failure

In this study, mortality was 4.8% while on the second line treatment which occurred within 5 days. This finding is similar to the previous study done in 2013 in Nagpur, India in which 4.9% died due to treatment failure [16]. However, our finding is high compared to the studies done in South Africa 2006, in which 2.2% died while on second line antibiotics treatment [24]. On the other hand, this finding is low compared to the studies done in India and Kenya 2009 and 2013 with mortality ranged from 10.2%– 16% respectively [9, 10]. In the study conducted in Kenya, children died within 5 days due to treatment failure at 48 hours [10]. The low death rate in our study could be attributed to the improved health services at our health facility as it is the tertiary referral hospital compared to the district hospital in a study done in Kenya.

Several limitations were faced during this study, including the inability to do blood culture to isolate the microorganism and to do drug susceptibility patterns for microorganisms. Also, we could not do sputum or bronchial secretion for gram stain, ZN stain, culture and sensitivity and viral antigen or serology test.

In conclusion, the study found that the proportion of children with first line antibiotics treatment (ampicillin and gentamicin) failure for SCAP was high and the risk of it was predicted by the presence of moderate and severe acute malnutrition, HIV infection, presentation with low oxygen saturation of <90%, convulsions, central cyanosis, and abnormal chest X-ray.

We therefore, recommend consideration of second line treatment and clinical trials for SCAP patients to reduce associated morbidity and mortality. A study to determine the aetiology of severe community acquired pneumonia is also highly recommended so that the susceptibility pattern of the first line antibiotics treatment can be established.

## Supporting information

**S1 File. This is the excel data spread sheet.**
(XLSX)

## Acknowledgments

The authors would like to thank all the children and Caregivers who volunteered to be enrolled in this study and Bugando Medical Centre pediatric nurses who assisted in the study in one way or the other.

## Author Contributions

**Data curation:** Restituta Phabian Muro.

**Formal analysis:** Tulla Sylvester Masoza, Benson R. Kidenya.

**Investigation:** Godfrey Kasanga.

**Methodology:** Benson R. Kidenya.

**Supervision:** Neema Kayange, Benson R. Kidenya.

**Writing – original draft:** Restituta Phabian Muro, Benson R. Kidenya.

**Writing – review & editing:** Tulla Sylvester Masoza.

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
