## [Decision Letter · Decision Letter 0]

18 Jun 2020

PONE-D-20-04002

Title - Predictors and outcome of first line treatment failure among underfive children with community acquired severe pneumonia at Bugando medical centre, Mwanza, Tanzania: A cross sectional study

PLOS ONE

Dear Dr. Masoza,

Thank you for submitting your manuscript to PLOS ONE. After careful consideration, we feel that it has merit but does not fully meet PLOS ONE’s publication criteria as it currently stands. Therefore, we invite you to submit a revised version of the manuscript that addresses the points raised during the review process.

We look forward to receiving your revised manuscript.

Kind regards,

Natasha McDonald

Associate Editor

PLOS ONE

Journal Requirements:

Reviewers' comments:

Reviewer's Responses to Questions

**Comments to the Author**

1. Is the manuscript technically sound, and do the data support the conclusions?

Reviewer #1: Partly

Reviewer #2: No

2. Has the statistical analysis been performed appropriately and rigorously? 

Reviewer #1: Yes

Reviewer #2: No

3. Have the authors made all data underlying the findings in their manuscript fully available?

Reviewer #1: Yes

Reviewer #2: No

4. Is the manuscript presented in an intelligible fashion and written in standard English?

Reviewer #1: Yes

Reviewer #2: Yes

5. Review Comments to the Author

Reviewer #1: The authors present very interesting data on first-line antibiotic resistance for severe pneumonia and associated predictive factors.

In my opinion, however, the discussion of the results should be oriented differently. The authors should evaluate these results in the light of the current WHO guidelines, not only those dedicated to severe pneumonia, but also to HIV / AIDS and severe malnutrition. The shocking incidence rate of resistance observed by the study (>50%), deserves a wider discussion, also in consideration of the fact that the strong impact of severe malnutrition and HIV / AIDS infection on this pathology is very well known.

I strongly recommend reviewing the whole discussion in this perspective. The work thus conceived would contain an element of important novelty, capable of distinguishing it from the many papers dedicated to the subject.

Reviewer #2: Pneumonia remains a leading cause of child mortality in LMICs. In the last decade there have been changes in WHO syndrome pneumonia definitions, altered aetiology because of conjugate vaccine introduction and an emergence of antimicrobial resistance. Hence, rigorous data on treatment outcomes are potentially valuable. This manuscript is interesting and clearly written but lacks some key details.

This seems to be a cohort study with follow up during 7 days rather than a cross sectional study as per the title.

It is absolutely essential in the methods to precisely define SCAP. A standardised definition, ideally as per the latest WHO syndrome definition, rather than individual clinician judgement is necessary in order to interpret the findings and make comparisons with published data from other sites.

Typically, selecting variables from univariate analyses to take into a multivariable analysis uses P values larger than 0.05, such as 0.1 in order to accommodate variables that may become significant after adjusted for confounding effects. The authors may wish to consider estimating relative risk rather than odds ratio since associations with events in the future (e.g. after 48h) are being tested.

In results, please list the reasons for exclusion with numbers of children for ‘had other diagnoses’. This could be a supplementary table if needed. This is important for comparison to other data.

In results, under participant’s characteristics, please define ‘respiratory distress’ and criteria for malnutrition.

Overall, this is a cohort of high severity children with very common hypoxia, lethargy etc, as the authors note in the Discussion. This may be related to participant selection (hence the need to precisely define SCAP).

What was the definition of treatment failure? Again, specific a priori criteria rather than clinicians judgement and individual behaviour are needed. For example, see https://www.ncbi.nlm.nih.gov/pmc/articles/PMC3691501/

Were any participants discharged prior to 48h or 7 days? If so, how was this handled in the analysis?

Were second line antibiotics ever given before 48h?

How many children died among those who were not changed to second-line treatment?

A flow chart would be helpful in communicating numbers that were enrolled, deaths before 7 days, failed at 48h, changed treatment, discharged or died between 48h and 7 days.

The Discussion section is difficult to assess without the definitions for inclusion and for treatment failure. For example, comparable enrolment criteria (severity) would be needed to propose that higher failure rates than other studies may be due to AMR.

The statement that ‘overwhelming majority of cases of SCAP are caused by bacterial disease’ is not correct. See the results of the PERCH global pneumonia aetiology study. Reference 12 is old (likely pre-conjugate vaccines) and found bacteraemia in only a minority (15.6%) of cases.

In conclusions, a clinical trial would be needed before recommending starting with second line antibiotics as evidence would be needed that it has better efficacy and what impact on AMR that expanded ceftriaxone use would have.

‘gentamycin’ is spelt ‘gentamicin’

6. PLOS authors have the option to publish the peer review history of their article (what does this mean?). If published, this will include your full peer review and any attached files.

Reviewer #1: No

Reviewer #2: Yes: Professor James A Berkley

---

## [Author Response · Author response to Decision Letter 0]

31 Jul 2020

Reviewer #1: 

The authors present very interesting data on first-line antibiotic resistance for severe pneumonia and associated predictive factors.

In my opinion, however, the discussion of the results should be oriented differently. The authors should evaluate these results in the light of the current WHO guidelines, not only those dedicated to severe pneumonia, but also to HIV / AIDS and severe malnutrition. The shocking incidence rate of resistance observed by the study (>50%), deserves a wider discussion, also in consideration of the fact that the strong impact of severe malnutrition and HIV / AIDS infection on this pathology is very well known.

I strongly recommend reviewing the whole discussion in this perspective. The work thus conceived would contain an element of important novelty, capable of distinguishing it from the many papers dedicated to the subject.

Response:

Thank you for the comment and we greatly appreciate the positive feedback. Authors have revised the discussion on the impact of HIV/AIDS and severe malnutrition on the response of first line antibiotic treatment for SCAP. See from line 296 to 312 of the revised manuscript with track change.

Reviewer #2: 

1. Pneumonia remains a leading cause of child mortality in LMICs. In the last decade there have been changes in WHO syndrome pneumonia definitions, altered aetiology because of conjugate vaccine introduction and an emergence of antimicrobial resistance. Hence, rigorous data on treatment outcomes are potentially valuable. This manuscript is interesting and clearly written but lacks some key details.

Response:

Thank you. We greatly appreciate your feedback. The key details missing have been added in the revised manuscript. 

2. This seems to be a cohort study with follow up during 7 days rather than a cross sectional study as per the title.

Response:

Thank you for this observation.. The authors agree that the manuscript presented for publication purely contains the prospective data since the outcome of interest was studied 48 hours post-enrollment of participants into the study. 

3. It is absolutely essential in the methods to precisely define SCAP. A standardised definition, ideally as per the latest WHO syndrome definition, rather than individual clinician judgment is necessary in order to interpret the findings and make comparisons with published data from other sites.

Response:

Thank you for this comment. We have incorporated the definition of SCAP applied at the time of the study as per WHO pocket book guideline of 2013. Please see line 109-117 of the revised manuscript with track changes

4. Typically, selecting variables from univariate analyses to take into a multivariable analysis uses P values larger than 0.05, such as 0.1 in order to accommodate variables that may become significant after adjusted for confounding effects. The authors may wish to consider estimating relative risk rather than odds ratio since associations with events in the future (e.g. after 48h) are being tested.

Response:

Thank you for this comment. After discussion and literature review (https://doi.org/10.1111/j.1553-2712.2010.00773.x) on use of OR vs RR in cohort studies, authors agreed to redo the analysis using generalized linear models to obtain RR since the incidence of outcome has shown to be common >10% (first line treatment failure of 50.6%), hence there are chances of OR to over-estimate the RR. Please see line 168 to 182 and line 229-238 of the revised manuscript with track changes.

5. In results, please list the reasons for exclusion with numbers of children for ‘had other diagnoses’. This could be a supplementary table if needed. This is important for comparison to other data.

Response:

Thank you for this comment. In our study, participants who had “Other diagnoses” were considered as not eligible for enrolment unless the participant also presented with features of SCAP

6. In results, under participant’s characteristics, please define ‘respiratory distress’ and criteria for malnutrition.

Response:

Thank you for this comment. We have defined respiratory distress and criteria for moderate and severe acute malnutrition. Please see line 202 to 209 of the revised manuscript with track changes.

7. Overall, this is a cohort of high severity children with very common hypoxia, lethargy etc, as the authors note in the Discussion. This may be related to participant selection (hence the need to precisely define SCAP).

Response:

Thank you for this comment. We think this cohort is of severe disease possibly because it has involved children who met criteria for a severe form of community-acquired pneumonia whom most of them get referred from regional and district hospitals to Bugando Medical Centre (study site) which is a tertiary level hospital. The definition for SCAP has now been included in the manuscript. Please see line 100 to105 and 109 to117 of the revised manuscript with track changes

8. What was the definition of treatment failure? Again, specific a priori criteria rather than clinician’s judgment and individual behaviour are needed. For example, see https://www.ncbi.nlm.nih.gov/pmc/articles/PMC3691501/

Response:

Thank you for this comment. We have now defined treatment failure under the material and methods section. Please see line 137 to 141 of the revised manuscript with track changes

9. Were any participants discharged prior to 48h or 7 days? If so, how was this handled in the analysis?

Response:

Thank you. No participant was discharged prior to 48 hours, but some were discharged at around day 5 of hospital stay. Since the primary outcome was first line treatment failure that was observed at 48 hours after enrolment, discharge at around day 5 of hospital stay did not affect the analysis of the desired outcome

10. Were second line antibiotics ever given before 48h?

Response:

Thank you. Yes there were children with SCAP who were initiated on the second line soon after admission regarding the attending clinician judgment, and this made them ineligible to the study

11. How many children died among those who were not changed to second-line treatment?

Response:

Thank you. No children among those who were not changed to the second line died in the 7 days follow up period

12. A flow chart would be helpful in communicating numbers that were enrolled, deaths before 7 days, failed at 48h, changed treatment, discharged or died between 48h and 7 days.

Response:

Thank you for this comment. A flowchart has been added as supporting information file. Please see lines 448 to 449 of the revised manuscript with track changes

13. The Discussion section is difficult to assess without the definitions for inclusion and for treatment failure. For example, comparable enrolment criteria (severity) would be needed to propose that higher failure rates than other studies may be due to AMR. 

Response:

Thank you for this comment. Most of the discussion has been revised and the missing definitions have been added as stated in the above response. Please see discussion from line 264 to 338 of the revised manuscript with tract changes.

14. The statement that ‘overwhelming majority of cases of SCAP are caused by bacterial disease’ is not correct. See the results of the PERCH global pneumonia aetiology study. Reference 12 is old (likely pre-conjugate vaccines) and found bacteraemia in only a minority (15.6%) of cases.

Response:

Thank you for this comment. The quoted statement was removed. Please see line 273 of the revised manuscript with track changes

15. In conclusions, a clinical trial would be needed before recommending starting with second line antibiotics as evidence would be needed that it has better efficacy and what impact on AMR that expanded ceftriaxone use would have.

Response:

Thank you for this comment. Authors agree that a clinical trial will put more light on the matter, but we think in a situation where there are comorbid such as HIV infection and severe malnutrition starting second line treatment would serve lives of these children. For example in our setting a study done among malnourished children on antimicrobial susceptibility found 100% resistance to Ampicillin among the gram-negative isolates. Hence adoptions of these findings into local guidelines are important. https://www.ncbi.nlm.nih.gov/pmc/articles/PMC5267369/

16. ‘gentamycin’ is spelt ‘gentamicin’ 

Response: Thank you for this comment. The misspelling has been rectified. Please see line 81, 267, 346 of the revised manuscript with track changes

---

## [Decision Letter · Decision Letter 1]

26 Aug 2020

PONE-D-20-04002R1

Title - Predictors and outcome of first line treatment failure among under-five children with community acquired severe pneumonia at Bugando medical centre, Mwanza, Tanzania: A prospective cohort study

PLOS ONE

Dear Dr. Masoza,

Thank you for submitting your manuscript to PLOS ONE. After careful consideration, we feel that it has merit but does not fully meet PLOS ONE’s publication criteria as it currently stands. Therefore, we invite you to submit a revised version of the manuscript that addresses the points raised during the review process.

The reviewers regarded your revision as considerably improved and with potential to inform the topic area. The two main issues raised by reviewer 2: a) to indicate deaths before 48h and b) give the number excluded because of second line treatment, and a careful spelling check need to be addressed. Minor edits as per the other comments, including moving methods out of the results section and considering the inclusion criteria for the prior studies discussed would also help improve the manuscript.

---

## [Author Response · Author response to Decision Letter 1]

8 Oct 2020

1. The two main issues raised by reviewer 2:

 a) To indicate deaths before 48h 

Response:

 No death before 48 hrs of follow up 

Kindly Refer Study flow chart

b) Give the number excluded because of second line treatment

Response:

718 children not eligible for enrolment

• 416 children were out of study age limit

• 269 children had no features of SCAP

• 33 children had SCAP but were started on second line antibiotic treatment

Kindly Refer Study flow chart

2. A careful spelling check need to be addressed. 

Response:

 Spelling check was done.

3. Minor edits as per the other comments, including moving methods out of the results section 

Response:

 Summary of enrolment process was merged with participants’ characteristics

4. Considering the inclusion criteria for the prior studies discussed would also help improve the manuscript.

Response:

Thank you for this comment. The authors could not review the inclusion criteria since it would alter the current study findings, and may alter the conclusion drawn.

---

## [Editor Report · Decision Letter 2]

27 Oct 2020

PONE-D-20-04002R2

Title - Predictors and outcome of first line treatment failure among under-five children with community acquired severe pneumonia at Bugando Medical Centre, Mwanza, Tanzania: A prospective cohort study

PLOS ONE

Dear Dr. Masoza,

Thank you for submitting your manuscript to PLOS ONE. After careful consideration, we feel that it has merit but does not fully meet PLOS ONE’s publication criteria as it currently stands. Therefore, we invite you to submit a revised version of the manuscript that addresses the points raised during the review process.

We look forward to receiving your revised manuscript.

Kind regards,

James A Berkley

Academic Editor

PLOS ONE

Additional Editor Comments (if provided):

Thank you for these minor revisions which clarify entry to the study. One further minor revision would make this valuable paper more rigorous and useful to policymakers. The abstract concludes that 2nd line antimicrobials should be considered by clinicians, however as stated in the discussion, we do not know if this would improve outcomes - no information on this is provided by this study and it needs a clinical trial. I suggest: "Conclusion: Half of the children with SCAP at this tertiary center had first line antibiotics treatment failure. HIV infection and acute malnutrition were independently predictive of first line treatment failure. We recommend consideration of second line treatment and clinical trials patients for SCAP to reduce associated morbidity and mortality."

---

## [Author Response · Author response to Decision Letter 2]

21 Nov 2020

Response to reviewers

Comment

Thank you for these minor revisions which clarify entry to the study. One further minor revision would make this valuable paper more rigorous and useful to policymakers. The abstract concludes that 2nd line antimicrobials should be considered by clinicians, however as stated in the discussion, we do not know if this would improve outcomes - no information on this is provided by this study and it needs a clinical trial. I suggest: "Conclusion: Half of the children with SCAP at this tertiary center had first line antibiotics treatment failure. HIV infection and acute malnutrition were independently predictive of first line treatment failure. We recommend consideration of second line treatment and clinical trials patients for SCAP to reduce associated morbidity and mortality."

Response

Thank you for the suggestion of the conclusion. The conclusion in the abstract and at the end of the discussion was amended. Please see line 47-51, and 306 to 316 of the revised manuscript with track changes.

---

## [Editor Report · Decision Letter 3]

25 Nov 2020

Title - Predictors and outcome of first line treatment failure among under-five children with community acquired severe pneumonia at Bugando Medical Centre, Mwanza, Tanzania: A prospective cohort study

PONE-D-20-04002R3

Dear Dr. Masoza,

We’re pleased to inform you that your manuscript has been judged scientifically suitable for publication and will be formally accepted for publication once it meets all outstanding technical requirements.

Kind regards,

James A Berkley

Guest Editor

PLOS ONE

Additional Editor Comments (optional):

Thank you, the requested change highlighting the need for clinical trials has been made.
---

## [Editor Report · Acceptance letter]

1 Dec 2020

PONE-D-20-04002R3 

Predictors and outcome of first line treatment failure among under-five children with community acquired severe pneumonia at Bugando Medical Centre, Mwanza, Tanzania: A prospective cohort study 

Dear Dr. Masoza:

I'm pleased to inform you that your manuscript has been deemed suitable for publication in PLOS ONE. Congratulations! Your manuscript is now with our production department. 

Kind regards, 

on behalf of

Dr. James A Berkley 

Guest Editor

PLOS ONE